# Myeloproliferative Neoplasms: Contemporary Review and Molecular Landscape

**DOI:** 10.3390/ijms242417383

**Published:** 2023-12-12

**Authors:** Muftah Mahmud, Swati Vasireddy, Krisstina Gowin, Akshay Amaraneni

**Affiliations:** 1Department of Medicine, Midwestern University Internal Medicine Residency Consortium, Cottonwood, AZ 86326, USA; 2Department of Medicine, University of Arizona Health Sciences, Tucson, AZ 85701, USA; 3Division of Hematology and Oncology, Department of Medicine, University of Arizona Cancer Center, Tucson, AZ 85701, USA

**Keywords:** primary myelofibrosis, secondary myelofibrosis, MDS-MPN, molecular and mutational landscape

## Abstract

Myelofibrosis (MF), Myeloproliferative neoplasms (MPNs), and MDS/MPN overlap syndromes have a broad range of clinical presentations and molecular abnormalities, making their diagnosis and classification complex. This paper reviews molecular aberration, epigenetic modifications, chromosomal anomalies, and their interactions with cellular and other immune mechanisms in the manifestations of these disease spectra, clinical features, classification, and treatment modalities. The advent of new-generation sequencing has broadened the understanding of the genetic factors involved. However, while great strides have been made in the pharmacological treatment of these diseases, treatment of advanced disease remains hematopoietic stem cell transplant.

## 1. Introduction and Brief Epidemiology

Myelofibrosis was first reported more than 140 years ago by Gustav Heuck in 1879. However, it was not until 1951 that the term ‘myeloproliferative neoplasms’ was conceived by William Dameshek, who is now known for the creation of the concept of myeloproliferative disorders [1,2]. Myelofibrosis (MF) is a Philadelphia (*BCR-ABL1*)-negative myeloproliferative neoplasm (MPN) characterized by bone marrow fibrosis, extramedullary hematopoiesis, leucoerythroblastic, and organomegaly. Its main biological feature is noted to be the overactivity of the Janus kinase-signal transducer and activator of the transcription (JAK-STAT) signaling pathway, consisting of somatic mutations involving Janus kinase 2 (JAK2), calreticulin (CALR), and myeloproliferative leukemia virus (MPL) genes, which together constitute 90% of the driver mutations [3,4].

Primary myelofibrosis (PMF) is divided into two phases by the international consensus classification (ICC): “pre-fibrotic” and “overtly fibrotic”. In addition, about 15% of patients with essential thrombocythemia (ET) or polycythemia vera (PV) develop a PMF-like phenotype over time, a phenomenon referred to as secondary myelofibrosis (SMF), post-ET myelofibrosis, or post-PV myelofibrosis (MF). These syndromes have a similar treatment and outcome to PMF [5].

Myelofibrosis has considerable biological and clinical heterogeneity. It is broadly divided into the myelodepletive (or cytopenic) phenotype, and the myeloproliferative phenotype as illustrated by Figure 1 below. The former is highly prevalent in PMF and is characterized by low blood counts. The latter is typically associated with SMF, mild anemia, a minimal need for transfusion support, and normal to mild thrombocytopenia. Differences in somatic driver mutations and allelic burden, as well as the acquisition of non-driver mutations like additional sex combs like transcriptional regulator 1 (ASXL)1, serine/arginine-rich splicing factor 2 (SRSF2), and U2 small nuclear RNA(U2AF1), further affect these phenotypic variations, prognoses, and responses to therapies such as JAK2 inhibitors [6].

In the United States, the prevalence of MF is estimated to be between 4 and 6 per 100,000 people, with a median survival of 2 to 11 years in primary MF patients [7,8]. Prevalence rate increases with age, with some preponderance in the Ashkenazi population.

## 2. Molecular and Mutational Landscape

### 2.1. Role of Driver Mutations

In the last decade, there has been a seminal change in the molecular landscape of myelofibrosis. This epoch started with the 2005 discovery of the role of the JAK-STAT pathway in MPN. A somatic mutation in exon 14 of the JAK2 gene l, characterized by a valine substitution to phenylalanine in position 617, results in a conformational change in the domain of the JH2 pseudokinase of JAK2. This leads to constitutive activation of the JAK2-driven signal pathway in the absence of binding the Erythropoietin receptor (EPOR), MPL, and G-CSFR ligands [9,10]. In addition, mutated CALR and MPL can activate the JAK-STAT signaling pathway. Collectively, they converge on a final pathway, with JAK activation determining the downstream phosphorylation of STATs. This results in the dimerization and transfer of STATs to the nucleus to activate or suppress gene transcription, causing cell proliferation and survival of relevant myeloid lineage cells [11]. 

Mutations in JAK2, CALR, and MPL are known altogether as the driver mutations. Of the three driving genes, the JAK2 V617F mutation occurs in 60% of PMF cases (Figure 2), while the CALR mutation occurs in 20–30% of cases (with predominance in Type 1/1 similar mutations), and the MPLW515L/K mutation occurs in 5–10% of cases [6].

The role of JAK2 V617F in MPN is explained by mitotic recombination, which causes a copy-neutral loss of heterozygosity (LOH) along a variable region on 9p, which can be observed in both ET and PV [12]. However, PV has a higher variant allelic frequency (VAF) at 50% than ET at 25%, and higher than post-ET or post-PV myelofibrosis at 100%. This progressive increase in VAF from PMF to SMF represents clonal dominance [12,13]. CALR mutations and MPL mutations affect 30% and 10% of SMF, respectively [13].

The pathogenic role of the driver mutation in myelofibrosis is partly explained by its phenotypic manifestation. JAK2V617F alleles are associated with older age, high hemoglobin levels, high leukocyte count, low platelet counts, and a higher risk of thrombosis. On the contrary, JAK2V617F VAF below 25% is associated with characteristics of a cytopenic MF phenotype, including low leukocyte count and low hemoglobin, rather than a myeloproliferative MF phenotype. It also represents an independent predictor of an increased mortality rate in patients with PMF [14]. This further reflects the clinical relevance of the JAK2 allelic burden [15]. In contrast to the JAK2 mutations, patients with CALR-mutated PMF were shown to be younger, to have higher platelet counts, were less likely to require blood transfusions for anemia or to present with leukocytosis, and to have a generally more indolent clinical course with better overall survival than JAK2- or MPL-mutated cases [16,17].

The PMF category devoid of any of the three canonical driver mutations is classified as “triple-negative” (TN), which represents between 10 and 15% of PMF patients (Figure 2) [18]. These pathogenetic differences have prognostic implications as PMF being TN is an independent variable for transformation to acute myeloid leukemia and a reduction in survival [19,20]. Despite the interest in myelofibrosis and the ever-expanding fund of knowledge, information on TN SMF remains limited.

### 2.2. Role of Non-Driver Mutations 

The emergence and rapid adoption of next-generation sequencing (NGS) has changed the understanding of how other genetic modifiers contribute to the etiopathogenesis of myelofibrosis. Non-driver mutations are a spectrum of deleterious genetic alterations in DNA methylation, epigenetic regulators, spliceosomes, oncogenes, and transcription factors all contributing to disease progression and leukemic transformation [21]. Mutations in epigenetic regulators are common in PMF, and a targeted sequencing study by Teferi et al. revealed mutations in 81% of these patients as opposed to in 50% of ET and PV patients [22,23]. These additional mutations, also known as cooperating mutations, increased with the progression of the disease [24].

These mutations include DNA methylation regulators TET methylcytosine dioxygenase 2, DNA methyltransferase 3A, isocitrate dehydrogenase 1/2 (TET2, DNMT3A, IDH1/2), histone/chromatin modulators (ASXL1, EZH2), splicing factors (SRSF2, SF3B1, U2AF1, ZRSR2), signal transduction factors (SH2B3/LNK, CBL), and transcription factors (TP53 and RUNX1) [22,23]. However, the most common mutations are ASXL1 (36%), TET2 (18%), SRSF2 (18%), and U2AF1 (16%), with 35%, 26%, 10%, and 9% of patients, respectively, carrying one, two, three, or greater than four mutations [22]. ASXL1 mutations result in loss of PRC2-mediated histone H3 tri-methylation and repression of known leukemogenic target genes [25]. All ASXL1 mutations have been shown to have an equipotent adverse prognostic impact in PMF [26,27]. They are among the high-risk mutations (HMRs) which also include EZH2 (present in 4–7% of patients), IDH1 and IDH2 (1–3%), SRSF2 (8–15%), and U2AF1 [28]. The HMR mutation reduces overall survival (OS) and leukemia-free survival (LFS) in patients with pre-fibrotic and overt PMF, a phenomenon exacerbated by the presence of more than one HMR mutation [29]. The role of ASXL1 and HMRs in SMF is uncertain [30]. The summary of these mutations is shown in the Table 1 below.

Interestingly, spliceosomes and splice site mutations are important non-driver or cooperating mutations in myelofibrosis. Most observed mutations are heterozygous missense mutations that lead to a modified function and can affect the RNA splicing, ultimately resulting in the accumulation of functionally defective proteins, which phenotypically leads to cytopenia [12]. The genes involved include SF3B1, SRSF2, and U2AF1. SRSF2 is a commonly mutated splicing factor observed in 18% of PMF patients. It clusters with IDH mutations and is an independent predictor of reduced survival [31]. SFB3BI occurs in less than 10% of MF but in more than 80% of PMF where it plays a greater role in disease progression to myelofibrosis. Overall, while it may not affect disease outcome and phenotype in MF, it has been shown to be associated with transfusion dependency [32]. Mutation U2AF1 involved in splice-site recognition was associated with poor overall survival and relapse-free survival in patients receiving allogeneic hematopoietic stem cell transplantation for MF [33]. Collectively, mutations in these genes are associated with AML transformation [12].

### 2.3. Role of Megakaryocytes

Megakaryocytes are derived from hematopoietic stem cells (HSCs) through a thrombopoietin (TPO)-dependent process controlled by various transcription factors including, but not limited to, the Runt-domain (RD) transcription factors X1 (RUNX1), GATA binding protein 1 (GATA1), and growth factor independent 1B (GFI1B [34]).

Furthermore, megakaryocytes play a dual role in maintaining HSC dormancy during homeostasis and in promoting their regeneration after chemotherapy. This is accomplished through an array of cytokines and immunomodulators such as megakaryocyte-derived transforming growth factor-β (TGF-β), C-X-C motif ligand 4 (CXCL4), and fibroblast growth factor-1 (FGF1) [35].

In MF, megakaryocytes typically demonstrate a variety of morphologic abnormalities, which have been seen to play a direct role in PMF via proximity to other HSCs. They also have a direct impact on the HSC cycle through cytokine secretion, promotion of bone marrow fibrosis, and induction of small antitumor molecules by megakaryocyte maturation as seen in MPN models [36].

Megakaryocytes also express driver mutations similar to other HSC types. This is reflected in their early procurement in a primitive hematopoietic cell with subsequent progression from ET, PV, or pre-PMF into overt PMF or acute leukemia after the acquisition of additional somatic or non-driver mutations [20,23,37,38]. It has also been shown that megakaryocytes in PMF exhibit defects in maturation associated with reduced GATA1 protein [39].

### 2.4. Role of Endothelial Cells

A theory postulated with an important clinical implication given the propensity for vascular complications in this patient population in the molecular landscape of MPN is the role of EC- and CD34-positive hematopoietic stem cells (HPSCs) in MF; the concurrent presence of driver mutations was demonstrated in both cell lines in a cohort of 14 patients, with the homology possibly affecting disease outcomes and progression [40].

### 2.5. Role of Innate and Acquired Immunity

Immune dysregulation is a pathological feature of MPNs that supports clonal evolution, mediates symptoms, and leads to immunosuppression [41]. Genetic and cellular changes that define cancer provide the immune system with the ability to produce T-cell responses that recognize and eliminate cancer cells [42]. The suppression of intrinsic and acquired immune surveillance mechanisms and the destruction of the tumor-immunity cycle enable the expansion of neoplastic myeloid clones. The reduced ability of antigen-presenting cells (APCs) in patients with myelofibrosis and other MPNs to process antigens leads to a reduction in the priming and activation of T cells [41].

In addition, there is a downregulation of human leukocyte antigen (HLA) genes and major histocompatibility complex (MHC) class I and II genes [43]. Romano et al. reported a decreased pool of circulating dendritic cells (DCs) with defective monocyte differentiation and a reduction in the circulating T-helpers (Th)1 and Th17, together with suboptimal innate lymphoid cells (ILCs) in MF patients with JAK2, CLR, and triple negative mutations [44].

The modified immune landscape of MPNs results in increased susceptibility to serious and life-threatening fungal, viral, and bacterial infections. The highest risk was observed in myelofibrosis compared to ET or PV (hazard ratio 3.7 vs. 1.7) [45]. As a result of increased predisposition to infection in MPN and an elevated risk of developing serious complications, most guidelines recommend that patients with MPN receive annual influenza and other inactivated vaccines, despite diminished response to vaccination [46]. They are advised against live attenuated vaccines [41].

MPN driver mutations lead to the activation of a pro-inflammatory signaling cascade, which includes tumor necrosis factor (TNF)/nuclear factor kappa-light-chain-enhancer of activated B cells (NF-kB) pathway in mutated HSCs and their progeny [47]. This phenomenon causes cell-extrinsic effects leading to chronic inflammation with the perturbation of the bone marrow microenvironment, which contributes to the MPN phenotype and renders the niche less supportive of normal hematopoiesis in a self-reinforcing cyclical manner [48].

## 3. Clinical Features of Myelofibrosis

The underlying pathology and phenotypic manifestations drive the clinical features of the disease. These include severe anemia, marked liver hepatosplenomegaly, constitutional symptoms (e.g., fatigue, night sweats, fever), weight loss, bone pain, splenic infarction, pruritus, thrombotic symptoms, and bleeding tendencies [49].

The most common presenting symptom of these is severe fatigue in 50 to 70 percent of patients [50]. Splenomegaly results from extramedullary hemopoiesis and is noted in 25 to 50 percent. Hepatomegaly is noted in up to 40 to 70 percent of patients [50] with the downstream complications of portal hypertension and variceal bleeding including non-hepatosplenic extramedullary hematopoiesis that might lead to cord compression, pleural effusion, pulmonary hypertension, and widespread musculoskeletal pain [51]. The observed anemia is generally secondary to ineffective erythropoiesis.

The incidence of arterial and veinous thrombosis in PMF (2 patients per 100 years) is about the same as in ET (1 to 3 patients per 100 years) and is significantly lower than in polycythemia vera (5.5 patients per 100 years) [52].

The most common causes of death in myelofibrosis include the progression of leukemia in about 20% of patients, followed by other comorbidities, including cardiovascular diseases, and cytopenic effects, including infection and bleeding [49,53].

## 4. MDS/MPN

Myelodysplastic syndrome (MDS)/myeloproliferative neoplasm (MPN) overlap syndromes are a spectrum of chronic clonal myeloid malignancies that share features of MDS and MPN at the time of presentation [54]. Though they exhibit different morphological diagnostic features, they have similar disease phenotypes such as cytopenia, susceptibility to acute myeloid leukemia, and significantly reduced survival [55].

The intersection of MDS and MPN characteristics (Figure 3) in these conditions is identified by the occurrence of cytopenia due to dysplasia and of elevated blood cell counts due to myeloproliferation. These can exist concurrently or sequentially, and the overlap occurs at all levels including in clinical features, pathogenesis, molecular landscape, and treatment options [56]. This issue is exacerbated by morphology changes and the evolution of dysplastic features throughout disease progression even in traditional MPN. These obstacles require more objective methods of diagnosis, eventually leading to the identification of some of the abnormalities that triggered their pathogenesis.

In 2022, the World Health Organization (WHO) classified MDS/MPN neoplasms into five classes with four distinct adult varieties and one pediatric entity. These varieties are chronic myelomonocytic leukemia (CMML); MDS/MPN with neutrophilia (formerly atypical chronic myeloid leukemia or aCML); MDS/MPN with SF3B1 mutation and thrombocytosis (formerly MDS/MPN-RS-T); MDS/MPN not otherwise specified (formerly MDS/MPN-U); and juvenile myelomonocytic leukemia (JMML) [57,58]. Due to obscure clinical presentation, underdiagnosis, and previous difficulties in classification, precise global prevalence is unknown. However, incidence rates reported in the United States per 100,000 people are as follows: 0.6 for CMML (0.57–0.63); 0.06 for aCML (0.04–0.62); 0.07 for MDS/MPN-U (0.006–0.009); MDS/MPN-RS-T < 1% of new MDS cases; and 1.2 for JMML. CMML is the most common [59].

The pathophysiology of MDS/MPN results from abnormalities in the regulation of myeloid pathways for cellular proliferation, maturation, and survival. This results in a clinically variable syndrome involving a combination of both driver and non-driver mutations, though none are essential for causation. The precise etiology remains unknown.

## 5. Molecular and Mutational Landscape of MDS/MPNs

Diagnosing MDS/MPNs is challenging due to shared features with other myeloid neoplasms, and diagnosis frequently requires expert pathologic review in addition to careful clinical correlation. It requires the absence of chromosomal rearrangements such as PDGFRA, PDGFRB, FGFR1, BCR-ABL1, or PCM1-JAK2 fusion [60], which are associated with any known diseases. This is in addition to other WHO and ICC classification criteria.

The somatic mutations involved in MDS/MPNs play a significant role in the disease biology, clinical presentation, and prognosis. They may offer a therapeutic window of opportunity by identifying druggable targets, especially with next-generation sequencing becoming a standard. These mutations may involve signal transduction pathways, RNA splicing, DNA transcription and repair, and other epigenetic mutations. However, there is no specific defining mutation.

CMML is the most common of MDS/MPNs with a characteristic hallmark of high frequency in somatic mutations in TET2, SRSF2, and ASXL, with an estimated 60%, 50%, and 40%, respectively [61,62]. TET2 mutations occur early in the course of the disease, and the clonal dominance established is responsible for the monocytic phenotype [63]. It has also been found that approximately 40% of patients have mutations in their cell signaling pathways, specifically in the RAS pathway (NRAS, KRAS, CBL) and JAK2. These mutations are more commonly observed in the proliferative subtype [64].

Atypical chronic myeloid leukemia (aCML) is a rare BCR-ABL1-negative MDS/MPN characterized by myeloid immaturity, including hyperplastic myeloid hyperplasia and peripheral blood leukocytosis [65]. It has heterogeneous clinical presentation and genetic features, a high rate of transformation to acute myeloid leukemia, and dismal survival outcomes.

Recurrent molecular abnormalities most frequently noted are SET binding protein 1 (SETBP1), affecting approximately 25–40% of patients, and ethanolamine kinase 1 (ETNK1), affecting approximately 8% of patients [66,67]. Furthermore, aCML patients often have mutations like ASXL1, EZH2, TET2, SRSF2, and N/KRAS, also found in other MDS/MPNs. These mutations are present in over 20% of aCML cases [62,66]. The presence of JAK2 V617F in aCML is uncommon, occurring only in 3% to 7% of cases. Its presence and similar BCR-ABL1-negative MPN-associated mutations in CALR and MPL may indicate an alternate diagnosis within the relevant clinicopathologic context, such as PV, ET, or myelofibrosis [57,67]. The previously associated CSF3R is now shown to occur in a relatively lower frequency.

MDS/MPN-RS-T had its nomenclature recently updated in the 2022 WHO classification to MDS/MPN with SF3B1 mutation and thrombocytosis. This represents the intersection of clinicopathologic features of the subgroup of MDS with ring sideroblasts (MDS-RS) with MPNs, such as essential thrombocytosis. Spliceosome mutations SF3BI are genes involved in RNA splicing, protein synthesis, and mitochondrial function, suggesting common mechanisms of action in MDS, while mutation in JAK2 is responsible for the MPN components. The spliceosome mutation is the initial mutation in the disease. Almost all cases of wild-type SF3B1 display co-mutation of JAK2 and ASXL mutations in addition to the alternative spliceosome gene. In a small number of cases, there are cytogenetic abnormalities which include trisomy 8 among other complex and monosomal karyotypes [68].

JMML is a rare but heterogenous myeloid malignancy with similar clinical and molecular features to CMML [60]. It is characterized by hypersensitivity to granulocyte-macrophage colony-stimulating factor (GM-CSF) caused in 85–90% of cases by somatic or germline mutation of a gene involving the RAS-MAPK pathway [69,70,71]. These mutations are usually observed in PTPN1, followed by N/KRAS, NF,1 and CBL [71,72]. Of note, this entity was re-classified into myeloproliferative disorders by the WHO in 2022.

MDS/MPN-U, now known as MDS/MPN NOS in the 5th edition of the WHO 2022 classification, is a diagnosis of exclusion given to patients who exhibit dysplastic and proliferative features but do not meet the criteria for a defined MDS/MPN. It shows an overlap with other MDS/MPN entities. Notable mutations in this cohort are ASXL1 observed in 29–56% of cases and TP53 mutations in 8–9% of cases, the latter being associated with poor survival outcomes. These are higher frequencies than what is observed in other MDS/MPNs with other common mutations including SRSF2, SETBP1, JAK2V617F, NRAS, and TET2 [73,74,75]. Cytogenic abnormalities noted include trisomy 8, monosomy 7, deletion 7q, and deletion 20q. The classifications were summarized in Table 2 below.

### Differences in the Mutational Landscape of MDS/MPN and MF/MPN

There are similarities between MDS/MPN and MF in terms of clinical heterogeneity, morphologic presentation, and risk for eventual transformation to AML. However, there are also distinct differences. The three driver mutations in the majority of MF/MPNs represent a unique molecular signature and causal association. The reverse is true for MDS/MPNs.

In a large metanalysis of 53 studies by Wan et al., ASXL site mutations were found to be more frequent in MF than in MDS/MPN [78]. Additionally, the TET2 mutation was found to confer a favorable prognosis in CMML [79], but in MF/MPF, it was associated with an increased risk of thrombosis [80] and monocytic differentiation with a comparatively smaller prognostic significance in survival [81].

Cytogenetic abnormalities play a more significant role in MDS/MPNs compared to MF/MPNs. This is shown by the new provisional creation of a new sub-entity MDS/MPN with isolated isochromosome (17q) categorized under MDS/MPN NOS in the ICC classification [82], which has a unique preservation of the TP53 allele. These cases were found to have shorter medial overall survival [83]. Isochromosome 17q abnormalities do exist in MF/MPN as part of non-driver mutations and are of prognostic relevance but do not exist as a distinct clinicopathologic entity.

## 6. Therapeutic Overview and Treatment Paradigm

Myeloproliferative neoplasms are chronic medical conditions with sequential but unequal risks of progression to acute leukemia, a phenomenon referred to as blast phase or leukemic transformation. The natural history starts from genetic defects in early childhood with clinical studies using whole-genome sequencing revealing that the time between acquiring somatic mutations and clinical manifestation can be prolonged, taking decades and resulting in substantial tumor heterogeneity [84,85].

Depending on the number of blasts and on the disease spectrum, the MPN can be divided into chronic phase disease (MPN-CP) with less than 10% blast in the bone marrow, and advanced disease phenotypes. Advanced disease is stratified functionally into the MPN-accelerated phase (MPN-AP) with 10 to 19% blasts, and the MPN-blast phase (MPN-BP), with greater than 20% of blasts as highlighted in the Table 3 below:

### 6.1. Treatment of Chronic Phase MPN 

Disease-related thrombotic and hemorrhagic complications, as well as the risk of blast transformation, are responsible for the majority of morbidity and mortality seen in this subgroup. Management focuses on the prevention of these complications and impeding the progression of disease. This is typically achieved through a multifaceted approach beginning with an appropriate risk stratification model, encompassing age (less than or equal to 60 years), thrombotic syndromes for ET and PV, lifestyle, and prophylactic intervention with ASA. For MF, risk-profiling tools include Mutation-and Karyotype-Enhanced International Prognostic Scoring System (MIPSS 70) and Genetically Inspired Prognostic Scoring System (GIPSS) [51]. Others include Dynamic International Prognostic Scoring System (DIPSS), Myelofibrosis Secondary to Polycythemia and Thrombocythemia-Prognostic Model (MYSEC), and International Prognostic Scoring System (IPSS). 

Despite the ability to more accurately score and identify these diseases, existing medications have limited effectiveness in modifying the course, and their toxicity profiles often prevent prolonged use, ultimately doing little to prevent leukemic transformation. This has led to a concerted effort to address this unmet pharmacologic need.

### 6.2. Role of Cytoreductive Agents

Hydroxycarbamide or hydroxyurea (Table 4) is the initial cytoreductive agent of choice in MPN-CP, typically encompassed by ET and PV. It inhibits DNA synthesis by blocking ribonucleotide conversion and acts predominantly in the S phase of the cell cycle [86]. Hydroxyurea is better than phlebotomies in preventing cardiovascular events and myelofibrosis progression according to early reports from the Polycythemia Vera Study Group in the 1980s [87]. It also has an additional utility in treating MPN-associated splenomegaly, as has been demonstrated in various small-sized trials. Efficacy in ET has also been established.

However, prolonged use of hydroxyurea has been associated with intolerance, treatment-related toxicities, and resistance in 15–24% of patients in a large-cohort 890-patient population [88]. Resistance is defined by ongoing requirement for phlebotomy to maintain hematocrit <45, worsening myeloproliferation (WBC > 10, PLT > 450), and hematological toxicity at minimum therapeutic doses (PLT < 50, ANC < 1.0) [89]. 

Patient populations with a TP53 mutation and spliceosome site mutations are at relatively higher risk of hydroxyurea resistance in contrast to patients with a heterozygous JAK 2 mutation. Hydroxyurea resistance is a negative prognostic disease marker, and these patients have an increased risk of thrombosis, transformation, and poor survival outcomes [88,90,91]. This resistance to hydroxyurea drove the search for other alternatives.

One alternative is anagrelide, a potent platelet inhibitor used in the treatment of ET and MPN. It is an amidazoquinazolin compound initially designed as an anticoagulant with strong platelet-reducing effects [92]. It is a platelet-specific cytoreductive agent which inhibits megakaryocyte colony development and maturation with additional upstream activity against GATA1 and Friend of GATA1 (FOG 1 [93,94]).

### 6.3. Role of Interferons

Interferons (IFNs) are a group of immunomodulatory cytokines with a wide range of activity and additional functions as a cancer defense mechanism (Table 4). They have long been known to have activity against MPN. However, poor pharmacokinetics and the adverse effect profile of earlier forms limit their usage [95]. Progressive modification of the recombinant product has prompted renewed interest in recent years, with several regulatory approvals and other products in the pipeline. The exact mechanism of action is unknown but has been postulated to be by activation of a PKCδ-ULK1-p38 MAPK signaling cascade [96]. 

Results from the PROUD PV trial showed that ropeginterferon alfa-2b achieved the primary endpoint of a complete hematological response (CHR) and was non-inferior to hydroxyurea [97]. The effectiveness of ropeginterferon alfa-2b was sustained beyond the 36-month mark compared to hydroxyurea in the CONTINUATION-PV trial. This success is largely attributable to the ability of the newer molecule to circumvent the pharmacokinetic limitation of the earlier versions of IFN.

### 6.4. Role of JAK Kinase Inhibitors

The thematic role of the JAK2V61F mutation in the pathogenesis of MPN has provided a druggable target used in the management of this disease. The demonstration of the effectiveness and improved outcomes with ruxolitinib (JAK1/JAK 2 inhibitor) in the COMFORT trial enabled it to become the standard of care [98]. Since then, the group has expanded to include fedratinib as well as pacritinib–fedratinib (Table 4) which is a combination molecule of JAK2/FLT3 inhibitor approved in 2019 and useful either as a frontline therapy or second-line agent in the setting of ruxolitinib failure [99]. Pacritinib is a JAK2- and IRAK1-specific inhibitor indicated for the treatment of myelofibrosis in patients with severe thrombocytopenia [100].

Resistance to JAK inhibitors is poorly understood and is an unmet clinical need. One of the mechanisms cited is the persistence of the MAPK pathway, including MEK and ERK kinases. Therefore, these must be targeted along with JAK2 inhibitors for better therapeutic results [101].

### 6.5. Accelerated MPN, Blast Phase, and Leukemic Transformation

MPN-AP and MPN-BP represent the progression of the disease from a steady state in a select patient population, with MPN-AP being an intermediary step with a blast count from 10 to 19%. The transformation to blast phase is known as leukemic transformation and represents more than 20% of blasts in bone marrow or peripheral blood [102]. This indicates the final end-product of various epigenetic modifications leading to self-sustaining proliferative signaling, successful immune surveillance escape, replicative immortality, and metastatic potential [103]. This portends a prognosis worse than de novo or primary leukemia.

All three driver mutations contribute to this process, with the JAK/STAT pathway being the most predominant. Other important predisposing non-driver mutations with a role in disease outcome include TET2, EZH2, and TP53 [104]. The pattern of genetic aberrations observed in post MPN-AML include IDH1/2, TET2, ASXL1, EZH2, and SRSF2, while FLT3, NPM1, and DNMT3 are implicated in de novo disease [105].

### 6.6. Treatment of Accelerated MPN and MPN Blast Phase

Treatment of MPN AP/BP is centered around HSCT, as it is the only curative treatment associated with the most leukemic-free survival. This was demonstrated by Alchalby et al. in a large cohort of patients from a European working group, which showed a cure rate of 25% with 3-year progression-free survival and overall survival rates of 26% and 36%, respectively [106]. 

The successful outcome of hematopoietic stem cell transplantation (HSCT) in this patient population depends on a range of factors, including disease-specific prognostic factors (DPSs), the patient-specific comorbidity index, timing, induction regimen, and donor selection, thus requiring a multidisciplinary approach and collaboration [107].

However, other agents are used as a bridge to transplant. These include AML-based induction chemotherapy, hypomethylating agents, BCL2 inhibitor (venetoclax), and IDH1 and IDH2 inhibitors (ivosidenib and enasidenib).

### 6.7. Future Therapeutic Landscape

TP53 germline mutation and other genomic aberrations are found in up to 30% of MPNs, especially MPN-BP at the time of leukemic transformation, and this explains their presence in the dominant AML clone at high VAFS (>50%) [104]. This mutation is a negative prognostic factor associated with resistance to traditional chemotherapeutic regimens. However, the current understanding of the role of the overexpression and amplification of MDM2 and MDM4 in TP53 inhibition (through ubiquitination) is now being exploited as a therapeutic target. 

Nutlins are a class of medications that inhibit interaction between MDM2 and P53, leading to preferential non-genotoxic activation [104]. Candidates in development include navtemedlin and idasanutlin, currently in phase II trials.

In the setting of dysfunctional TP53, the G2/M checkpoint becomes crucial in preventing genomic instability. Wee1 inhibitors can cause cell death by overriding the G2/M checkpoint in such cells [108]. A combination of Wee1 inhibitor and PARP inhibitor has shown a decreased leukemic burden in mice models, providing a future potential target [109].

Research indicates that inflammation in the bone marrow microenvironment as well as systemic inflammation contribute to the development and progression of MPNs, with compounding effects of driver mutation leading to the creation of an inflammatory milieu further supporting the proliferation of aberrant cells [110]. The overexpression of TGF-B, IL2, and IL8 has prognostic implications, with IL8 overexpression associated with increased mortality, circulating blasts, and reduced leukemia free survival [111]. However, there is no ongoing trial targeting this process.

Other novel agents in various phases of clinical trials include magrolimab, a monoclonal antibody against CD-47 signal regulatory protein (SIRPα) interaction with the downstream effect of enhancing macrophage activity to destroy tumor cells. Eprenetapopt, a small-molecule P53 reactivator, is currently undergoing phase III trials with use in combination with azacytidine following favorable phase II results showing relapse-free survival at one year. 

## 7. Conclusions

MF, MPN, and MDS/MPN overlap syndromes represent a complex group of hematological disorders with a diverse spectrum of clinical presentations and molecular abnormalities, making their diagnosis and classification challenging. Advances in next-generation sequencing have illuminated these genetic aberrations and allowed for a deeper understanding of disease mechanisms. With this comes the ability to predict the risk of disease, such as the presence of TET2, SRSF2, and ASXL mutations pointing toward the development of CMML, or the poor prognosis of MDS/MPN NOS associated with TP53 mutations. It also allows for the ability to compare the same gene across diseases, contrasting TET2′s favorable prognosis in CMML with its propensity for increased risk of thrombosis in MF, or the involvement of JAK-STAT in MPN with its almost complete absence in aCML. Further stratification allows for an in-depth understanding of chronic and acute phases. As our knowledge of the underlying molecular mechanisms continues to evolve, so does the potential for more targeted and effective treatments, from interferons and checkpoint inhibitors to promising new small molecule reactivators. Further research and collaboration are essential to unraveling the intricacies of these disorders and improving the lives of affected individuals.

## Figures and Tables

**Figure 1 ijms-24-17383-f001:**
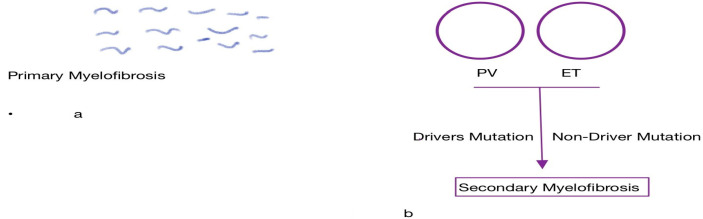
Primary (**a**) and secondary myelofibrosis (**b**).

**Figure 2 ijms-24-17383-f002:**
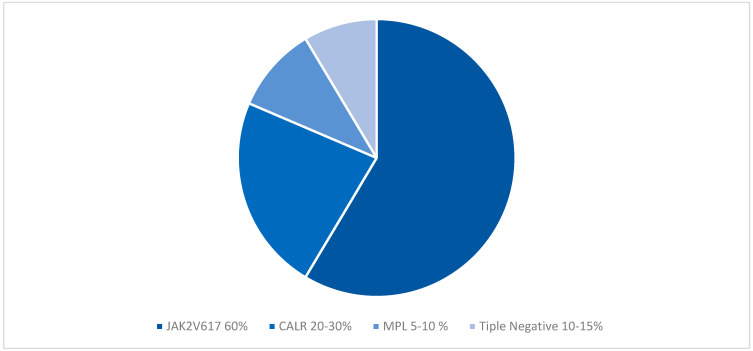
Distribution of Major Driver Mutation and Triple Negative Disease.

**Figure 3 ijms-24-17383-f003:**
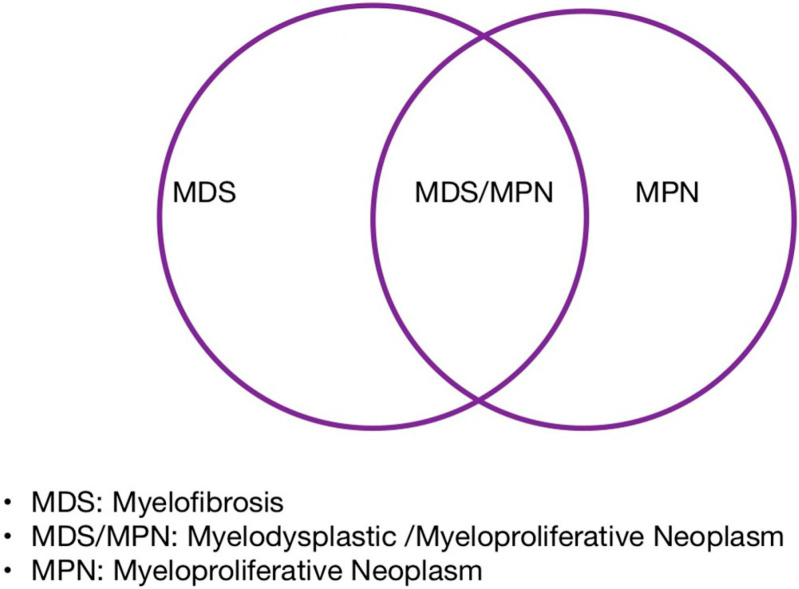
Intersectional relationship of MDS/MPN.

**Table 1 ijms-24-17383-t001:** Most common non-driver mutations in myelofibrosis in a sample of 147 patients [22].

**Mutation Category**	**Relative Frequency (%)**
ASXL1	36
TET2	18
SRSF2	18
U2AF1	16

**Table 2 ijms-24-17383-t002:** MDS/MPN classification: WHO 2022 nomenclature.

**Class**	**Frequencies of Common Somatic Mutations**	**Cytogenetic Abnormalities**	**Incidence Rate in the US** **/100,000**
Chronic myelomonocytic leukemia	TET2 (60%), SRSF2 (50%), ASXL (40%), NKRAS and JAK2 (40%)	Trisomy 8 (+8), −Y, (monosomy 7 and del7q), trisomy 21 (+21) [76]	0.6 (0.57–0.63)
Atypical chronic myeloid leukemia (aCML)	SETBP 1 (25–40%), ETNK1 (8%), ASXL1, EZH2, TET2, SRSF2, and N/KRAS	Trisomy 8, del(20q), −7/7q-, or isochromosome 17q [i17[q)] [77]	0.060 (04–0.62)
MDS/MPN with SF3B1 mutation and thrombocytosis	SFB3 (89.2%), JAK 2,	Trisomy 8	Less than 1% of all new MDS/MPN cases
MDS/MPN NOS	ASXL1 (29–56%), TP53 (8–9%), SRSF2, SETBP1, JAK2V617F, NRAS, and TET2	Trisomy 8, monosomy 7, deletion 7q, and deletion 20q.	0.07 (0.006–0.009)

Note: Mutations are not mutually exclusive. Relative percentages were added when available.

**Table 3 ijms-24-17383-t003:** Phenotypic staging of myeloproliferative neoplasm.

**Phenotypic Stage**	**Blast Percentage**	**Comment**
MPN CP	Less than 10% of blasts	May have a static course or progress to advanced disease depending on genetic risk profile
MPN AP	10 to 19% of blasts	
MPN BP	Greater than 20% of blasts	

**Table 4 ijms-24-17383-t004:** Therapeutic options for myeloproliferative neoplasms.

**Class**	**Examples**
JAK kinase inhibitors	Ruxolitinib, fedratinib
Combined JAK 2 and FLT3 inhibitors	Pacritinib–fedratinib
Cytoreductive agents	Hydroxyurea
Interferons	Ropeginterferon alfa-2b

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
