# Peer review of "Myeloproliferative Neoplasms: Contemporary Review and Molecular Landscape"

_ijms, 2023, doi:10.3390/ijms242417383_

Round 1

Reviewer 1 Report

Comments and Suggestions for Authors

Muftah Mahmud and colleagues have presented an interesting review on the molecular landscape of MPN and the impact of the combination of different mutations/alterations on the clinical manifestations and the therapeutic strategies.

The review is well written and structured. Despite that, it is full of many info and data, so I have some suggestions in order to improve the readability of the manuscript.

- I strongly suggest a Table summarizing the non-driver mutations in the section 2.2. Among this part, a comment about the recently reported  identification of mutations shared between CD34 and other cell populations in patients affected with PMF (PMID 34685741).

- I suggest a Table reporting the alterations, the diagnosis and the incidence of the lesions reported in the section number 5

- I think that an image or a scheme will improve the comprehension of the therapies and the related targets reported in the last part of the review. There are different approaches and a scheme or an image will strongly clarify the different fronts that are targetable at the moment.

- In the section concerning the therapy, a comment on the recent revision of the literature about the transplantation approach will be appreciate (PMID 35159362)

I hope my suggestions will help the authors in improving the quality and readabilty of the manuscript.

Comments on the Quality of English Language

English language is fine. Just some sentences to be rephrased.

Author Response

 Thank you for your insightful suggestions and valuable feedback.

  • We added a table summarizing the non-driver mutations in section 2.2.
  • We included a comment in section 2 of the manuscript regarding the shared mutations recently identified between CD34 and other cell populations in patients with PMF (PMID 35159362).
  • One of the changes made was including a Table 3 which reports classification, common mutations, cytogenetic abnormalities, and incidence of MDS/MPN.
  • The authors added a simplified scheme of representing various therapeutic approaches in section 6 (Fig 4)
  • The excellent observations made about the transplantation approach in (PMID 35159362) was added to the manuscript in section 6.

Reviewer 2 Report

Comments and Suggestions for Authors

The review “Myeloproliferative Neoplasms: Contemporary Review and Molecular Landscape” by Muftah Mahmud et al., is a very informative piece updating our knowledge about the myeloproliferative neoplasms. The review succinctly describe the epidemiology and the molecular features of the diverse types of myeloproliferative neoplasms. The topic is introduced well and the authors have provided a detailed description of the molecular mechanisms delineated in recent analyses. The various driver and non-driver mutations that the authors describe to cause or facilitate myeloproliferative neoplasms are interesting targets for future therapy. Also, the authors give a detailed account of the therapeutic/treatment possibilities that are currently in practice also helps to understand the huge diversity within the myeloproliferative neoplasms. The review correctly underscores the importance to understand the molecular landscape that leads to one or the other type of myeloproliferative neoplasm. Overall, it is an informative review. But, the review is text heavy and is without any diagrams or illustrations, which makes it a bit drab. Non-specialist readers will find it a bit disinteresting to read. Making illustrations of some of the molecular actors in diverse myeloproliferative neoplasms will make the reading more informative and easy to understand. Also, the authors need to expand the abbreviations at the point of first use such as JAK-STAT and ICC etc., and give the gene names followed by the symbols such as CALR, MPL, ASXL1, SRSF2 and U2AF1 etc. Incorporating these changes will make the review more relevant to the readers.

Author Response

These are excellent points, and we thank you for this constructive feedback. A total of 7 figures (Three tables and four illustrations) were added in relevant sections to improve readability and comprehension. We also expanded abbreviations at the point of first use where applicable.

Round 2

Reviewer 1 Report

Comments and Suggestions for Authors

The Authors strongly improved the quality of the manuscript.

I have some difficulties in understanding the figures the Authors added. Probably there are some errors or the loading procedures delated some details. Please, revise all the Figures.

Author Response

Dear Editors and Reviewers,

Find below the responses to the reviewers’ comments for the manuscript Myeloproliferative Neoplasms: Contemporary Review and Molecular Landscape, originally submitted to the MDPS Journal on 9/15/2023. The responses are highlighted in blue both here and in the manuscript.

Reviewer # 1

The Authors strongly improved the quality of the manuscript.

I have some difficulties in understanding the figures the Authors added. Probably there are some errors, or the loading procedures delated some details. Please, revise all the Figures.

Response:

Thank you for your positive feedback on the improved manuscript quality. All figures underwent a two-stage QC process; tables were reformatted, and notes were added where applicable. Some details may have been affected by loading procedures.

Thank you,

Authors.

Round 3

Reviewer 1 Report

Comments and Suggestions for Authors

I attached an example of figures that are present in the PDF file of the manuscript.

I think that something is wrong. If not, what is the meaning of the rectangles? What do the purple points represent? The same for all other figures. There are 2 or 3 rectangles in a couple of them. A figure like that has no sense. It is better to add no figure or a table.

Author Response

Dear Editors and Reviewers,

Find below the responses to the reviewers’ comments for the manuscript Myeloproliferative Neoplasms: Contemporary Review and Molecular Landscape, originally submitted to the MDPS Journal on 9/15/2023. The responses are highlighted in blue both here and in the manuscript.

Reviewer # 1

Comment about figures and color schemes; and recommendation for more conventional approach

Response:

Thank you for your constructive feedback. The authors changed Figure 2 to a pie chart and Figure 4 to a table.

The authors added a supplemental page with tables and figures per the editor’s request.

Thank you,

Authors.

Round 4

Reviewer 1 Report

Comments and Suggestions for Authors

The Authors have really improved the quality of the Figures.

Well done.